# Acetate-Induced Milk Fat Synthesis Is Associated with Activation of the mTOR Signaling Pathway in Bovine Mammary Epithelial Cells

**DOI:** 10.3390/ani12192616

**Published:** 2022-09-29

**Authors:** Miao Lin, Maocheng Jiang, Tianyu Yang, Dejin Tan, Guanghui Hu, Guoqi Zhao, Kang Zhan

**Affiliations:** 1Institute of Animal Culture Collection and Application, College of Animal Science and Technology, Yangzhou University, Yangzhou 225009, China; 2Institutes of Agricultural Science and Technology Development, Yangzhou University, Yangzhou 225009, China; 3Joint International Research Laboratory of Agriculture and Agri-Product Safety, The Ministry of Education of China, Yangzhou University, Yangzhou 225009, China

**Keywords:** bovine mammary epithelial cells, mTOR signaling pathway, acetate, milk fat, lipogenesis

## Abstract

**Simple Summary:**

The mechanistic target of the rapamycin (mTOR) pathway plays a vital role in promoting lipogenesis. Acetate induces de novo lipogenesis of fatty acids to synthesize milk fat in bovine mammary epithelial cells (BMECs). We hypothesized that acetate can enhance the expression of lipogenic genes and triglyceride (TG) production by activating the mTOR signaling pathway in BMECs. These results showed that TG synthesis was elevated (*p* < 0.01) in BMECs with acetate treatment, and the fatty acid profile in BMECs treated with acetate was affected. The mRNA levels of genes involved in the lipogenesis were upregulated (*p* < 0.05) in BMECs with acetate. Remarkably, expression of acetyl-CoA carboxylase α (ACCα) and fatty acid synthase (FAS) rate-limiting enzymes were upregulated in BMECs with acetate treatment. Moreover, the addition of acetate enhanced the protein expression of S6K1 kinase. In conclusion, these results suggest that the TG accumulation and expression of lipogenic genes induced by acetate are associated with the activation of the mTOR signaling pathway.

**Abstract:**

Acetate is a precursor substance for fatty acid synthesis in bovine mammary epithelial cells (BMECs), and the mTOR signaling pathway plays an important role in milk fat synthesis. However, the mechanism of the regulatory effects of acetate on lipogenic genes via the mTOR signaling pathway in BMEC remains unknown. We hypothesized that acetate can enhance the expression of lipogenic genes and triglyceride (TG) production by activating the mTOR signaling pathway in BMECs. Therefore, the aim of this study was to investigate the network of acetate-regulated lipid metabolism by the mTOR signaling pathway in BMECs. These results showed that TG synthesis was elevated (*p* < 0.01) in BMECs with acetate treatment. The lipid droplets were increased in the acetate-treated groups compared with those in the control group through the Bodipy staining of the lipids. In addition, the fatty acid profile in BMECs treated with acetate was affected, with an elevation in the proportions of C14:0, C16:0, and C18:0. The mRNA levels of the sterol-response-element-binding protein 1 (SREBP1), stearoyl-CoA desaturase 1 (SCD1), and fatty acid synthase (FAS) genes involved in the lipogenesis and transcriptional factors were upregulated (*p* < 0.05) in BMECs with acetate treatment. Remarkably, the expression of acetyl-CoA carboxylase α (ACCα) and FAS rate-limiting enzymes involved in lipogenesis was upregulated in BMECs with acetate treatment. Moreover, the addition of acetate enhanced the key protein expression of S6K1, which is related to the mTOR signaling pathway. Taken together, our data suggest that TG accumulation and expression of lipogenic genes induced by acetate are associated with the activation of the mTOR signaling pathway, which provides new insights into the understanding of the molecular mechanism in the expression of mTOR-signaling-pathway-regulated lipogenic genes.

## 1. Introduction

Of all of the components of milk, milk fat is mainly composed of triglycerides, and it is the key indicator of milk quality [1]. Moreover, milk fat reduces blood lipids, prevents cancer, and influences intestinal development [2,3]. Therefore, high-grade milk fat plays an important role in people’s lives. Half of the fatty acids in milk are long-chain fatty acids, which are directly transported from the blood [4]. The other half of the FAs in milk are synthesized de novo, with acetate being the primary precursor substrate [5,6].

It has been shown that acetate promotes milk fat synthesis by increasing the amount of FA synthesis in a dose–response manner [7]. Specifically, it is a two-carbon donor for malonyl-CoA synthesis and NADPH synthesis via the isocitrate pathway [8]. Therefore, acetate plays a vital role in the synthesis of milk fat in dairy cows. A previous study showed that milk fat was enhanced by feeding cows with sodium acetate [9]. However, the molecular mechanism by which acetate affects fat synthesis in BMECs remains unknown.

The mTOR signaling pathway is an axial pathway in the metabolism of nutrients. Recent studies have shown that the mTOR/S6K1 signaling pathway is activated by kisspeptin-10, which is a peptine hormone that further regulates the synthesis of milk fat [10]. Currently, SREBP-1c is dependent of the mTOR signaling pathway, which can control the expression of numerous genes involved in FA synthesis [11,12]. Acetate is mainly to utilized for de novo FA synthesis and further processing into TG. FA synthesis is catalyzed by acetyl-CoA carboxylase α (ACCα) and fatty acid synthase (FAS), which are involved in the key rate-limiting enzymes in lipid metabolism and are transcriptionally regulated by transcriptional regulators of sterol-response-element-binding protein 1 (SREBP1) in response to nutrients and hormones [13]. Rapamycin, a specific inhibitor of mTOR complex 1 (mTORC1), can block the expression of SREBP-1c target genes of FAS and stearoyl-coA desatase 1 (SCD1), suggesting that mTORC1 plays a role in FA biosynthesis [14,15]. However, the mechanism underlying the regulatory effects of acetate on the lipogenic genes via the mTOR signaling pathway in BMECs remains unknown.

Therefore, the aim of our study was to investigate the effects of acetate on the expression of genes related to milk fat synthesis and the mTOR signaling pathway in order to discuss these effects and their underlying mechanisms in milk fat synthesis.

## 2. Materials and Methods

### 2.1. Cell Culture

All operational procedures were carried out by following the guidelines of the Institutional Animal Care and Use Committee (IACUC) of Yang Zhou University. Mammary tissues were obtained from healthy lactating Holstein cows (day 100 of lactation), which were fed at the experimental farm of Yang Zhou University. The BMECs were obtained and cultured as previously described [16]. In brief, the culture medium was made up of Dulbecco’s Modified Eagle Medium/Nutrient Mixture F12 (DMEM/F12; Gibco, Shanghai, China) supplemented with 10% fetal bovine serum (FBS; Gibco, Shanghai, China), 100 U/mL penicillin, and 100 μg/mL streptomycin (Sigma-Aldrich, Shanghai, China). Then, cells were incubated at 37 °C in a 5% CO_2_ atmosphere.

### 2.2. Experimental Design

These BMECs were cultured in 6-well plates at a density of 2.5 × 10^5^ cells/well. The supernatant was removed after 12 h of incubation, and the cells were gently flushed with PBS three times. Then, these cells were stimulated with either DMEM/F12 medium (control group) or medium containing 12 mM acetate sodium (acetate group) for 6 d.

### 2.3. Determination of TG Content

After incubation for 6 days, these cells were obtained using trypsin and lysed for 10 min. Next, a triglyceride assay kit (Applygen, Beijing, China) was used to detect the levels of triglycerides in the BMEC lysis products. The results were normalized to the protein content.

### 2.4. Bodipy Staining

The BMECs were grown on a chamber slide to 70% confluency and subjected to incubation with 12 mM acetate for 6 days. After treatment, the cells were washed with PBS and fixed with 4% paraformaldehyde for 20 min at room temperature. After 3 washes with PBS, the cells were incubated with Bodipy 493/503 (1:1000 dilution; Amgicam, Shanghai, China) for 20 min at room temperature and then incubated with DAPI (Beyotime Biotechnology, Beijing, China) for 10 min. After extensive rinsing with PBS, the cells were observed with a confocal laser scanning microscope (Olympus, Tokyo, Japan).

### 2.5. RNA Isolation and Quantitative Real-Time PCR (qRT-PCR)

Total RNA was extracted from the BMECs with a TRIzol kit (Tiangen, Beijing, China). In order to determine the gene expression, the procedure was as follows: (1) 20 µL of reaction mixture was prepared with 1 μg of total RNA and 1× PrimeScript RT Master Mix (Takara, Beijing, China). (2) The reactions for performed for 15 min at 37 °C. (3) The system was heated to 85 °C for 5 s to inactivate the reverse transcriptase. (4) The program of qRT-PCR was at 95 °C for 5 s and 60 °C for 30 s with 40 cycles using SYBR^®^ Premix Ex TaqTM II (Takara). The relative gene expression of the genes was calculated with the GAPDH reference gene by using the 2^–ΔΔCT^ method. The primers for qRT-PCR are listed in the Table 1 [17].

### 2.6. Western Blotting

After incubation, RIPA lysis and an extraction buffer (Thermo Scientific, Shanghai, China) containing a 1× protease inhibitor cocktail (Thermo Scientific) and 1× phosphatase inhibitor cocktail tablets (Roche, Shanghai, China) were used for lysing and total protein extraction in the BMECs. The protein concentrations were determined using a BCA kit (Beyotime, Beijing, China). Equal amounts (40 µg) of protein lysates were fractionated by SDS-PAGE and transferred to nitrocellulose membranes (PALL, Shanghai, China). After being blocked with 5% horse serum, the membranes were incubated in Tris-buffered saline with Tween (TBS-T: 10 mM Tris–HCl, pH 7.5, 150 mM NaCl, 0.05% Tween 20) and the primary antibodies plus 5% horse serum with gentle shaking overnight at 4 °C. The following primary antibodies were obtained from Cell Signaling Technology (CST, Shanghai, China): GAPDH, mTOR, phosphorylated (p)-mTOR, S6K1, 4EBP1, p-4EBP1, ACCα, p-ACCα, FAS, and Perilipin-1 (1:1000). PPARγ (1:1000) was obtained from Abcam (Shanghai, China). The horseradish peroxidase (HRP)-conjugated secondary antibody was goat anti-rabbit IgG (1:5000; CST). The target bands were detected using the Super Signal West Femto Maximum Sensitivity Substrate or Pierce ECL Plus Western Blotting Substrate (Thermo Scientific).

### 2.7. Measurement of Fatty Acid in Bovine Mammary Epithelial Cells

The total lipids were extracted from BMECs in a 10 cm culture dish using 2 mL of 2.5% (vol/vol) sulfuric acid/methanol in a 15 mL glass tube with 200 μL of C11:0 fatty acid (Sigma, Shanghai, China) as an internal control. After ultrasonication for 10 min, the glass tubes were incubated at 80 °C for 60 min. After cooling the tubes to room temperature, 2 mL of 0.1 M HCl and 800 μL of hexyl hydride were added, and the glass tubes were vortexed for 30 s, followed by centrifugation for 5 min at 900× *g* and 4 °C. The supernatant was collected in a 2 mL silicified tube with 0.5 g of water-free sodium sulfate. After an acute shock, the tubes were centrifuged for 3 min at 13,800× *g* and 4 °C, and the liquid supernatant was collected and filtered with a 0.22 μm aqueous-phase membrane filter for fatty acid analysis through 7890A gas chromatography (Agilent Technologies, Shanghai, China) with an JW CP-Sil 88 (100 m × 0.25 mm × 0.20 μm, CP7489) column (Agilent Technologies, Shanghai, China). The amount of each fatty acid was determined. The results were normalized to the protein content.

### 2.8. Statistical Analysis

A statistical analysis was performed by using an independent sample t-test in the SPSS 19.0 software (SPSS Inc.; Chicago, IL, USA). *p* < 0.05 was considered significant, and <0.01 was considered highly significant.

## 3. Results

### 3.1. Effects of Acetate on TG Production in BMECs

The effects of acetate on the TG content in BMECs are presented in Figure 1. In response to acetate incubation, the level of TG was elevated (*p* < 0.05) compared with that in the control group. The lipid droplets were enriched in BMECs induced with acetate relative to those in the control group as a result of Bodipy staining of lipids (Figure 2).

### 3.2. Effects of Acetate on the Fatty Acid Composition in BMECs

Because of the increased expression of genes involved in TG synthesis, the cellular level of TG also increased in BMECs with acetate treatment. The fatty acid profile in BMECs treated with acetate showed profound changes. Consistently, C14:0, C16:0, and C18:0 were elevated (*p* < 0.05) in BMECs with the acetate treatment, whereas C13:0 and C18:1 were not affected (Table 2).

### 3.3. Expression of Genes Related to Lipid Metabolism in BMECs

SREBP-1, which acts as a transcription factor and controls lipid synthesis, was upregulated (*p* < 0.01) in BMECs treated with acetate (Figure 3A). Similarity, SCD1 and FAS, which were involved in lipogenesis, were elevated (*p* < 0.05) in BMECs incubated with acetate relative to the control group (Figure 3E,F). The other lipogenic genes showed no profound differences between the control group and acetate treatment group.

### 3.4. Effects of Acetate on the mTOR Signaling Pathway and the Expression of Proteins Involved in Lipid Metabolism in BMECs

The mTOR signaling pathway is associated with lipogenesis and the regulation of lipogenic genes. Compared with that in the control group, the protein expression of ACCα and FAS, which act as the rate-limiting enzymes for lipogenesis, was upregulated in BMECs with acetate treatment (Figure 4). Moreover, the addition of acetate enhanced the protein expression of S6K1, which is related to the mTOR signaling pathway (Figure 4). These results suggest that the expression of lipogenic enzymes induced with acetate is associated with the activation of mTOR signaling.

**Figure 3 animals-12-02616-f003:**
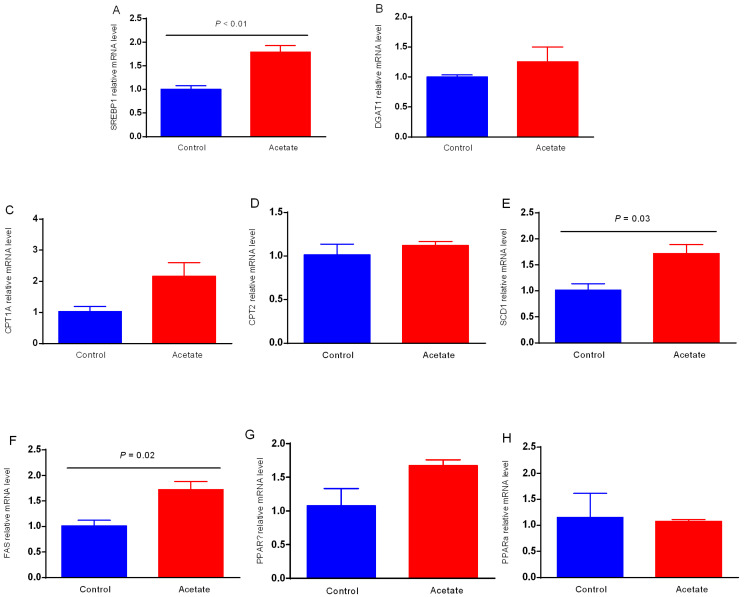
Effects of acetate on the mRNA levels of key genes involved in lipid metabolism. Bovine mammary epithelial cells were exposed to either 0 (as control group) or 12 mM acetate for the analysis of mRNA levels. Quantitative reverse-transcription PCR analysis of (**A**) sterol-response-element-binding protein 1 (SREBP1), (**B**) diacylglycerol O-Acyltransferase 1 (DGAT1), (**C**) carnitine Palmitoyltransferase 1A (CPT1A), (**D**) carnitine palmitoyltransferase 2 (CPT2), (**E**) peroxisome Stearoyl-CoA Desaturase 1 (SCD1), (**F**) fatty acid synthase (FAS), (**G**) peroxisome proliferatoractivated receptor-γ (PPARγ), and (**H**) peroxisome proliferatoractivated receptor-α (PPARα) in bovine mammary epithelial cells. GAPDH was used as an internal reference gene. The data are means ± SEM (n = 3).

## 4. Discussion

It is well known that acetate is the main lipogenic substrate in the induction of the process of de novo FA synthesis in BMECs. Our results demonstrate that the addition of acetate can enhance the TG production and FA synthesis in the mammary epithelia of dairy cows. This process involves the increased expression of ACC and FAS proteins that are related to key rate-limiting enzymes for lipogenesis via the activation of mTOR signaling. This study provides new insights into the understanding of the molecular mechanism underlying mTOR-signaling-pathway-regulated expression of lipogenic genes.

Milk fat is a component of the nutrient quality in milk, and it affects the economic value of dairy cows [6]. It has been reported that TG accounts for almost 97% of the milk fat content [18]. Acetate is the main carbon source in the carbon chain extension in the synthesis of fatty acids in milk. Kong et al. reported that acetate increased the concentration of TG in BMECs after 1 d of incubation [19]. In this study, we demonstrated that TG production and FA synthesis were elevated in BMECs with acetate treatment. Previous studies showed that the mTOR signaling pathway can regulate milk fat synthesis. However, the mechanism underlying the regulatory effects of acetate on the lipogenic genes via the mTOR signaling pathway in BMECs remains unknown.

The important role of mTOR in protein synthesis, lipid synthesis, and mitochondrial metabolism is often discussed [20]. In addition, S6K1 is an effector protein of the mTOR signaling pathway in cell growth, reproduction, and energy metabolism [21]. Previous studies have shown that the activated mTOR/S6K1 signaling pathway inhibits the pathogenesis of obesity. In this study, acetate was able to promote the S6K1 protein level in BMECs compared with that in the control group. In addition, acetate also induced the mRNA expression of the mTOR signal pathway’s downstream target gene, SREBP-1c. mTORC1 induces hepatic lipogenesis through the activation of the SREBP1 transcription factor in an S6K1-dependent manner [22], suggesting that S6K1 plays a vital role in SREBP1-mediated lipogenesis.

SREBPs, as a transcription factor family, promote the expression of genes related to FA synthesis and lipogenesis [23]. Specifically, SREBP1c can modulate many genes involved in de novo FA synthesis, which is necessary for milk fat synthesis in bovine mammary cells [24]. Recently, it was demonstrated in BMECs through gene overexpression and knockdown that SREBP1c is central in the regulation of the transcription of many genes involved in milk fat synthesis and secretion [25]. In this study, we found that the addition of acetate enhanced the gene expression of SREBP1c compared with the absence of acetate in BMECs. The SREBP-1c gene positively regulates the expression of ACCα, FAS, and SCD1 [26,27]. ACCα and FAS are key rate-limiting enzymes in de novo FA synthesis, and both of them are transcriptionally regulated by the transcriptional regulator of SREBP1. It has been demonstrated that expression of the FAS gene is associated with cellular lipid droplet formation in goat mammary epithelial cells [28]. In addition, the transcriptional regulation of milk fat synthesis during lactation in dairy cows involves FAS expression, which indicates that FAS is a vital element of milk fat synthesis [4]. Furthermore, the gene expression of SCD1 was upregulated in the peak lactation period, which was due to its function as a key enzyme in the synthesis of milk fat in mammary glands [4]. In SCD1-knockout goats, the milk fat percentage was reduced [29]. As expected, we found that the expression of ACCα, FAS, and SCD1 was upregulated with the addition of acetate. Therefore, TG production and the expression of lipogenic genes induced by acetate are associated with the activation of the mTOR–S6K1 signaling pathway, which provides new insights into the understanding of the molecular mechanism in the mTOR-signaling-pathway-regulated expression of lipogenic genes. These findings have an important significance in increasing milk fat and milk quality.

## 5. Conclusions

Acetate can enhance the TG production and FA synthesis in mammary epithelial cells of dairy cows. This process is dependent on the participation of the mTOR–S6K1 signaling pathway in the process of increasing the expression of ACCα and FAS proteins, which are related to the key rate-limiting enzymes for lipogenesis; S6K1 may be an important regulatory factor for the expression of lipogenic genes in BMECs.

## Figures and Tables

**Figure 1 animals-12-02616-f001:**
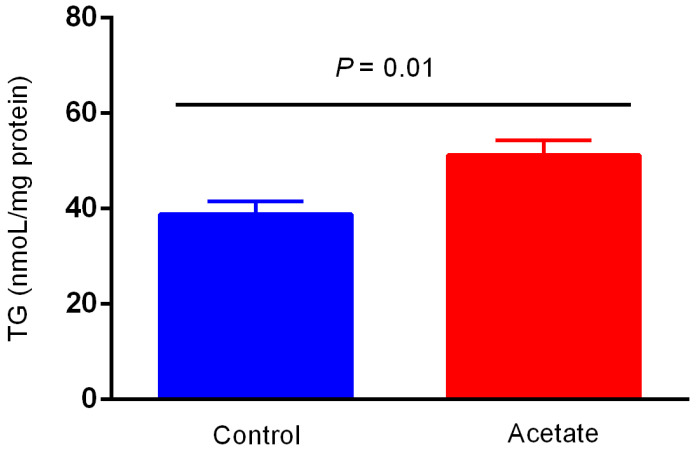
The effect of acetate on intracellular triacylglycerol (TG) production.

**Figure 2 animals-12-02616-f002:**
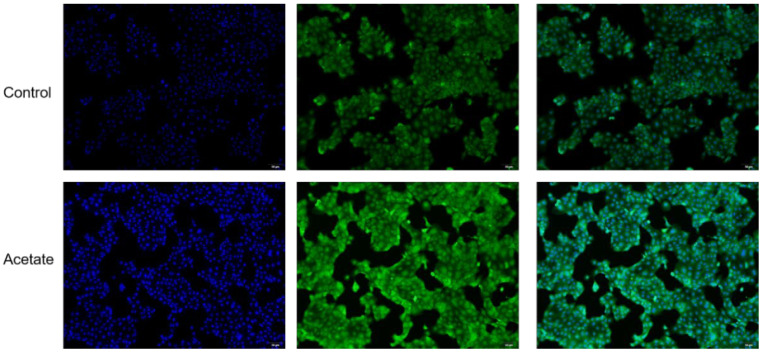
Bodipy staining showing lipid droplets in cultured BMECs induced with acetate. Lipid droplets were stained with Bodipy 493/503 (green), and nuclei were stained with DAPI (blue). Bar = 50 μm.

**Figure 4 animals-12-02616-f004:**
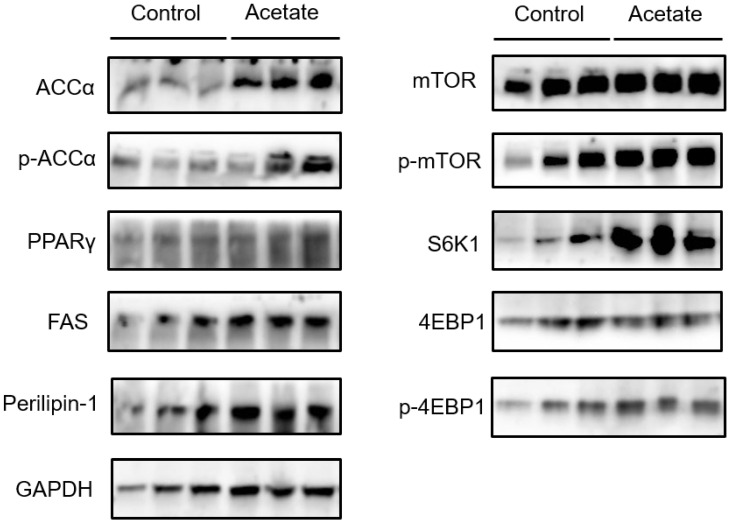
Effects of acetate on the expression of key proteins involved in the mTOR signaling pathway and lipid metabolism. Bovine mammary epithelial cells were exposed to either 0 (as a control group) or 12 mM acetate for a protein expression analysis. Western blotting analysis of mTOR, p-mTOR, S6K1, 4EBP1, p-4EBP1, ACCα, p-ACCα, FAS, perilipin-1, and PPARγ. The data were based on triplicate experiments.

**Table 1 animals-12-02616-t001:** Primer sequences used in this study.

Gene	Primer Sequence, 5′ to 3′	Source	Size (bp)
GAPDH	F: TTGTCTCCTGCGACTTCAACA R: TCGTACCAGAAATGAGCTTGAC	NM_001034034.2	103
DGAT1	F: GACACAGACAAGGACGGAGA R: CAGCATCACCACACACCAA	XM_025001414.1	141
CAPT1A	F: ACGCCGTGAAGTATAACCCT R: CCAAAAATCGCTTGTCCCTT	NM_001304989.2	119
CPT2	F: CACCATTAGAAGATACCTCAGTGC R: TCCAGTTTCAAAACTCTTACACAACT	XM_024985645.1	94
PPARA	F: TCAGATGGCTCCGTTATT R: CCCGCAGATCCTACACT	XM_005207472.4	112
PPARγ	F: CCAAATATCGGTGGGAGTCG R: ACAGCGAAGGGCTCACTCTC	NM_181024.2	101
SCD1	F: GGCACATCAACTTTACCACG R: CAGCCACTCTTGTAGCTTTCCTC	NM_173959.4	136
SREBP1	F: GACACCACCAGCATCAACCACG R: CAGCCCATTCATCAGCCAGACC	XM_024980343.1	117
FAS	F: ACAGCCTCTTCCTGTTTGACG R: CTCTGCACGATCAGCTCGAC	NM_001012669.1	144

**Table 2 animals-12-02616-t002:** The long-chain fatty acid composition of bovine mammary epithelial cells.

Item (μg/mg)	Control	Acetate	SEM	*p*-Value
C13:0	139	164	44	0.60
C14:0	482	926	157	0.048
C16:0	414	859	152	0.04
C18:0	459	880	123	0.03
C18:1	202	322	65	0.14

## Data Availability

Not applicable.

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
