# Peer review of "Acetate-Induced Milk Fat Synthesis Is Associated with Activation of the mTOR Signaling Pathway in Bovine Mammary Epithelial Cells"

_animals, 2022, doi:10.3390/ani12192616_

Round 1

Reviewer 1 Report

The manuscript entitled "Acetate-induced milk fat synthesis is associated with activation of mTOR signaling in bovine mammary epithelial cells" by Lin et al. to evaluate the effect of acetate on the milk fat synthesis and activation of mTOR signaling pathway in bovine mammary epithelial cells. Authors have some interesting finds that acetate can promote TG accumulation and expression of lipogenic genes (ACCα and FAS ) and activation of mTOR signaling pathway. However, specific comments need to be revised.

L49-50,delete the first sentence. It is a common issue and redundancy.

L96, 105, the authors should carefully review the typos throughout the text.

L 95, Grammar issue.

L134 Equal amounts of protein lysates were fractionated by SDS-PAGE. How much? Please add the amounts of protein lysates.

L143-155, Add temperature for centrifugation.

L154, Describe the procedure briefly, i.e. flowing gas, volume injected, flowing rate, temperature. Add fatty acid in detail in stead of “each fatty acid”. Why only test C13-C18 fatty acid?

L164, compared with control group

L166,185, Spelling mistake in “Control”.

L166,185, Figure 1 and 3 were presented in disunity formate.

L169-171, Figure 1 and 2 should be self-readable.

L 202, Figure 4, it is suggested to add gray analysis for the WB bands to make easily understand for these wb results.

L225-226, Delete.

Author Response

Reviewer 1

The manuscript entitled "Acetate-induced milk fat synthesis is associated with activation of mTOR signaling in bovine mammary epithelial cells" by Lin et al. to evaluate the effect of acetate on the milk fat synthesis and activation of mTOR signaling pathway in bovine mammary epithelial cells. Authors have some interesting finds that acetate can promote TG accumulation and expression of lipogenic genes (ACCα and FAS ) and activation of mTOR signaling pathway. However, specific comments need to be revised.

L49-50,delete the first sentence. It is a common issue and redundancy.

R: I have deleted the sentence.

L96, 105, the authors should carefully review the typos throughout the text.

R: I have rewritten the sentence.

L 95, Grammar issue.

R: I have rewritten the sentence.

L134 Equal amounts of protein lysates were fractionated by SDS-PAGE. How much? Please add the amounts of protein lysates.

R: I have added the amounts of protein lysates. Line 127

L143-155, Add temperature for centrifugation.

R: I have added temperature behind the speed. Line 145 and Line 147

L154, Describe the procedure briefly, i.e. flowing gas, volume injected, flowing rate, temperature. Add fatty acid in detail in stead of “each fatty acid”. Why only test C13-C18 fatty acid?

R: Because C13-C18 fatty acid can be detected in bovine mammary epithelial cells by 7890A gas chromatography (Agilent Technologies, Shanghai, China) using an JW CP-Sil 88 (100 m × 0.25 mm × 0.20 μm, CP7489) column (Agilent Technologies, Shanghai, China), but others types of fatty acid were not found in bovine mammary epithelial cells.

L164, compared with control group

R: I have added these words into the sentence. Line 158

L166,185, Spelling mistake in “Control”.

R: I have corrected the spelling.

L166,185, Figure 1 and 3 were presented in disunity formate.

R: I have revised them.

L169-171, Figure 1 and 2 should be self-readable.

R: I think that Figure 1 and 2 have been self-readable.

L 202, Figure 4, it is suggested to add gray analysis for the WB bands to make easily understand for these wb results.

R: Generally speaking, the gray analysis for the WB bands was performed by Image Soft, however, gray analysis was not actually accurate for the WB bands. Therefore, the WB bands do not need to be analyzed gray.

L225-226, Delete.

R: I have deleted the sentence. 

Reviewer 2 Report

The manuscript describes the effect of Acetate-induced milk fat synthesis is associated with activation of mTOR signaling pathway in bovine mammary epithelial cells.

The study was conducted correctly and provides understandable results.

The introduction describes a lot about the synthesis of fatty acids, but describes little about the effect of TOR on the synthesis of breast lipids, it needs to be modified in this regard.

The Materials and Methods are simple and therefore very legible.

The results are easy to read and the graphs and tables are very explanatory.

The discussion is correct but it is not precisely explained whether it is the TOR that activate the synthesis of lipogenetic genes or that it is acetate that is able to stimulate both.

However, with some more clarification on the mechanisms of functioning of the synthesis of mammary lipids, the manuscript can be published.

Author Response

Reviewer 2

The discussion is correct but it is not precisely explained whether it is the TOR that activate the synthesis of lipogenetic genes or that it is acetate that is able to stimulate both.

R: It is acetate that is able to stimulate both.

Reviewer 3 Report

Abstract.

The abstract is a bit on the lengthy side, it is up to the editors to decide on the matter.

Introduction.

Some of the text in the first two paragraphs contains basic information that can be found in textbooks. Everyone reading the article will be familiar with the topics. Hence, please reduced the length by deleting soma sentences.

The objectives of the study must be clearly described in a paragraph on their own.

Procedures.

Please describe the procedure for obtaining tissue samples

Was any histological assessment made to the tissue samples, to confirm that the mammary tissue was normal?

Table 1. Please add temperature and all other details of the PCR and please move to supplementary material.

t-test is the wrong test, please use correct statistical tests and redo the analyses.

Discussion.

Please add a paragraph about the clinical significance of the findings.

Conclusion

Please extend with another 1-2 sentences.

Author Response

Reviewer 3

Introduction.

Some of the text in the first two paragraphs contains basic information that can be found in textbooks. Everyone reading the article will be familiar with the topics. Hence, please reduced the length by deleting soma sentences. The objectives of the study must be clearly described in a paragraph on their own.

R: I have deleted some sentences, and the aims of the study is clearly described in a paragraph on their own.

Procedures.

Please describe the procedure for obtaining tissue samples

R: The mammary tissues were quickly excised and repeatedly rinsed using DMEM medium containing 500 U/mL penicillin, 500 µg/mL streptomycin, 250 µg/mL gentamicin, and 12.5 µg/mL amphotericin B (5×PSGA). Then, these mammary tissues were transported to laboratory in DMEM medium containing 5×PSGA on ice right now.

Was any histological assessment made to the tissue samples, to confirm that the mammary tissue was normal?

R: The mammary tissues were normal, because cows were free of clinical signs of disease.

Table 1. Please add temperature and all other details of the PCR and please move to supplementary material.

R: The PCR temperature was exhibited. In line 118

t-test is the wrong test, please use correct statistical tests and redo the analyses.

R: I am sure that t-test is right in comparison between two groups.

Discussion.

Please add a paragraph about the clinical significance of the findings.

R: I have added a paragraph. In line 251-252

Conclusion

Please extend with another 1-2 sentences.

R: I have added one sentence. In line 257-258

Round 2

Reviewer 3 Report

t-test is the wrong test, please use correct statistical tests and redo the analyses.

R: I am sure that t-test is right in comparison between two groups.

In relation to this dataset, analysis of variance will be the recommended test.
Please redo the analysis correctly.

Author Response

Dear editor,

The analysis of variance are consistent with present analysis of results by SPSS.